# Chromatin profiling in human neurons reveals aberrant roles for histone acetylation and BET family proteins in schizophrenia

Lorna A. Farrelly[1,12], Shuangping Zheng[2,12], Nadine Schrode[3], Aaron Topol[4], Natarajan V. Bhanu[5], Ryan M. Bastle[1], Aarthi Ramakrishnan[1], Jennifer C Chan[1], Bulent Cetin[1], Erin Flaherty[1,3], Li Shen [1], Kelly Gleason[6], Carol A. Tamminga [6], Benjamin A. Garcia[5], Haitao Li [2✉], Kristen J. Brennand [1,3,7,8,11✉] & Ian Maze [1,9,10✉]

Schizophrenia (SZ) is a psychiatric disorder with complex genetic risk dictated by interactions between hundreds of risk variants. Epigenetic factors, such as histone posttranslational modifications (PTMs), have been shown to play critical roles in many neurodevelopmental processes, and when perturbed may also contribute to the precipitation of disease. Here, we apply an unbiased proteomics approach to evaluate combinatorial histone PTMs in human induced pluripotent stem cell (hiPSC)-derived forebrain neurons from individuals with SZ. We observe hyperacetylation of H2A.Z and H4 in neurons derived from SZ cases, results that were confirmed in postmortem human brain. We demonstrate that the bromodomain and extraterminal (BET) protein, BRD4, is a bona fide 'reader' of H2A.Z acetylation, and further provide evidence that BET family protein inhibition ameliorates transcriptional abnormalities in patient-derived neurons. Thus, treatments aimed at alleviating BET protein interactions with hyperacetylated histones may aid in the prevention or treatment of SZ.

[1] Nash Family Department of Neuroscience, Friedman Brain Institute, Icahn School of Medicine at Mount Sinai, New York, NY 10029, USA. [2] Beijing Advanced Innovation Center for Structural Biology, MOE Key Laboratory of Protein Sciences, Department of Basic Medical Sciences, School of Medicine, Tsinghua University, 100084 Beijing, China. [3] Department of Genetics and Genomic Sciences, Pamela Sklar Division of Psychiatric Genomics, Icahn Institute of Genomics and Multiscale Biology, Icahn School of Medicine at Mount Sinai, New York, NY 10029, USA. [4] Graduate School of Biomedical Science, Icahn School of Medicine at Mount Sinai, New York, NY 10029, USA. [5] Epigenetics Institute, Department of Biochemistry and Biophysics, Perelman School of Medicine, University of Pennsylvania, Philadelphia, PA 19104, USA. [6] Department of Psychiatry, University of Texas Southwestern Medical School, Dallas, TX 75390, USA. [7] Black Family Stem Cell Institute, Icahn School of Medicine at Mount Sinai, New York, NY 10029, USA. [8] Department of Psychiatry, Icahn School of Medicine at Mount Sinai, New York, NY 10029, USA. [9] Department of Pharmacological Sciences, Icahn School of Medicine at Mount Sinai, New York, NY 10029, USA. [10] Howard Hughes Medical Institute, Icahn School of Medicine at Mount Sinai, New York, NY 10029, USA. [11] Present address: Departments of Psychiatry and Genetics, Wu Tsai Institute, Yale School of Medicine, New Haven, CT 065109, USA. [12] These authors contributed equally: Lorna A. Farrelly, Shuangping Zheng. ✉email: lht@tsinghua.edu.cn; kristen.brennand@yale.edu; ian.maze@mssm.edu

Schizophrenia (SZ) affects ~1% of the US population, and current treatments remain only partially efficacious, if at all, owing to our limited understanding of the pathophysiology and molecular underpinnings of the illness. This highly heritable genetic and neurodevelopmental disorder[1,2] arises as a result of complex interactions between hundreds of genetic (common and rare variants) and epigenetic factors (e.g., 3D genome architecture, histone posttranslational modifications/PTMs, chromatin "reader" proteins, etc.)[3,4]. Numerous studies to date have explored histone PTM landscapes in SZ brain and patient-derived cells using candidate-based approaches[5–13]. However, our understanding as to how these epigenetic mechanisms may contribute to disease etiology remains limited.

Owing to the impracticality of assessing live human brain tissues/cells in affected individuals, human-induced pluripotent stem cell (hiPSC)-derived neurons offer a physiologically relevant and readily accessible in vitro system for molecular analyses of "living" neurons within the context of disease. Here, employing unbiased histone PTM mass spectrometry, coupled to downstream mechanistic and functional validations in a well-characterized SZ hiPSC cohort[14], we identify aberrant interactions between SZ diagnosis and regulation of combinatorial hyperacetylated histone signatures [e.g., H2A.Z lysine 4, 7 and 11 acetylation (K4acK7acK11ac) and H4 lysine 5, 8 and 16 acetylation (H4K5acK8acK16ac)] in maturing neurons. Extending upon these findings, we then characterize the chromatin effector protein, BRD4, as a bona fide "reader" of H2A.Zac, and demonstrate that inhibition of BET family protein interactions with histone acetylation alleviates gene expression abnormalities in SZ neurons. As such, this work uncovers a previously uncharacterized epigenetic phenomenon in SZ neurons that may play an important role in disease etiology, and one that may be exploited in the development of future therapeutics aimed at alleviating symptoms of disease.

## Results

**Combinatorial histone hyperacetylation in SZ neurons**. Using unbiased, label-free mass spectrometry (LC-MS/MS), we performed combinatorial histone PTM assessments in hiPSCs, and hiPSC-derived neural progenitor cells (NPCs) and 4-week-old forebrain neurons derived from SZ cases vs. matched controls (Fig. 1a). Such analyses allow for simultaneous quantification of hundreds of histone PTM states within the same sample, both in isolation and in combination with adjacent PTMs (i.e., within peptide comparisons using bottom-up LC-MS/MS). Unexpectedly, while the field has focused primarily on classically permissive (e.g., H3K4me3, H3K27ac) vs. repressive (e.g., H3K27me3) PTMs in candidate-based assessments of histone regulation in SZ[5,15], we identified a unique pattern of combinatorial hyperacetylation occurring specifically on H2A.Z.1 (H2AZ) and H2A.Z.2 (H2AV) variant histone proteins (which differ by only three amino acids and are preferentially expressed in neurons vs. hiPSCs or NPCs; Supplementary Fig. 1a), as well as on H4, in SZ neurons relative to controls. While no differences in modification status were observed between SZ and controls in reprogrammed hiPSCs, elevated patterns of combinatorial acetylation (e.g., H2A.Z.1/2K4acK7acK11ac)—with concomitant loss of unmodified peptide signals—were found in SZ neurons, with more subtle increases also observed for single acetylation signatures on H2A.Z.2 in NPCs (Fig. 1b and Supplementary Data 1). Using site-specific antibodies that recognize combinatorially acetylated H2A.Z, such enhancements in global expression were further validated in both hiPSC-derived neurons and in postmortem dorsolateral prefrontal cortex (DLPFC) tissues from SZ cases vs. matched controls (Fig. 1c, d and Supplementary Table 1). Considering that the amino acid sequence of the N-terminal tail of histone H4 is strikingly similar to that of H2A.Z.1/2 proteins

(Supplementary Fig. 1b), H4 acetylation was also explored (Supplementary Fig. 1c). While more complex in its regulation, increased expression of combinatorial H4 acetylation (e.g., H4K5acK8acK16ac) was also observed in SZ NPCs and 4-week-old neurons, results that were validated via western blotting in hiPSC-derived neurons and postmortem DLPFC from SZ cases vs. matched controls (Supplementary Fig. 1d, e). In sum, these data indicate that unique combinatorial patterns of histone hyperacetylation occur in SZ neurons/brain tissues and may contribute, at least in part, to disease etiology.

**The chromatin "reader" protein BRD4, along with related BET family proteins, recognize combinatorial H2A.Z acetylation**. Next, to better mechanistically define what impact, if any, hyperacetylation of H2A.Z may have within SZ neurons, we turned our attention to investigations of chromatin effector proteins that may interact with these combinatorial states to influence phenotypic outcomes. One such "reader," the bromodomain containing protein BRD4, emerged as a likely candidate owing to its established binding interactions with H4 acetylation[16] and peptide sequence similarities that exist between H4 and H2A.Z N-termini.

BRD4 contains two bromodomains (Bromo1 and 2) that are known to recognize acetylated histones. Therefore, both domains were subsequently assessed for their binding to acetylated vs. unmodified H2A.Z.1 peptides (residues 1–15) (Fig. 2a). To directly quantify interactions between H2A.Z and the two bromodomains of BRD4, we performed isothermal titration calorimetry (ITC) to characterize their affinities to H2A.Z peptides bearing unmodified vs. di-acetylated (K4acK7ac) or tri-acetylated (K4acK7acK11ac) lysines. Consistent with our predictions, both Bromo1 and Bromo2 of BRD4 were found to recognize di- (Bromo1-$K_D$ = 4.81 mM; Bromo2-$K_D$ = 0.82 mM) and tri- (Bromo1-$K_D$ = 2.27 mM; Bromo2-$K_D$ = 0.60 mM) acetylated H2A.Z peptides in vitro, with an ~2.1 to 1.4-fold-binding preference, respectively, for tri- vs. di-acetylated H2A.Z (Fig. 2b, c). The binding of BRD4 to H2A.Zac is comparable to that of the Twist peptide (which contains a histone H4-mimic motif), in which binding affinities of 0.6 and 3 mM were measured for BRD4 Bromo1 and Bromo2, respectively[17]. Binding was not observed to the unmodified H2A.Z tail for either bromodomain tested.

To elucidate the molecular basis for acetylated H2A.Z readout by BRD4 Bromo2, we crystallized the Bromo2-H2A.Z (1–15) K4acK7acK11ac complex and determined its crystal structure at 1.5 Å resolution (Supplementary Table 2). In the complex structure, BRD4 Bromo2 displays a classical left-handed four-helical bundle fold, and the H2A.Z peptide is recognized through its canonical region comprising the ZA and BC loops (Fig. 2d). We were then able to model the H2A.Z (2–9) segment according to 2fo-fc omit electron densities (Fig. 2e). The K4 and K7 diacetyllysines were found to be inserted into an Asn-lined hydrophobic cage with K4ac anchored in the center. H2A.ZK11ac likely contributes to binding through additional contacts involving the acetyl group, however, this information is missing in the current structure, since crystal packing expelled segment H2A.Z (10–15) from binding. More specifically, the carbonyl oxygen of K4ac in H2A.Z forms a hydrogen bond with the side-chain amino group of N433 in BRD4 Bromo2 (Fig. 2f, i). K7ac in H2A.Z occupies the hydrophobic pocket consisting of the WPF shelf (W374, P375, F376) and V439 (Fig. 2f, ii). The carbonyl oxygen of A5 in H2A.Z is oriented towards the imido group of the H437 side-chain stabilized by a hydrogen-bond interaction (Fig. 2f, iii). According to the structural alignment of BRD4-Bromo1 (PDB: 4xy9) and Bromo2, the N433-P434-P435-E436-H437 in Bromo2 is replaced by N140-K141-P142-G143-D144 in Bromo1, which results in D144 in Bromo1 being too far apart to participate in hydrogen bonding with the K4 backbone carbonyl oxygen. This observation

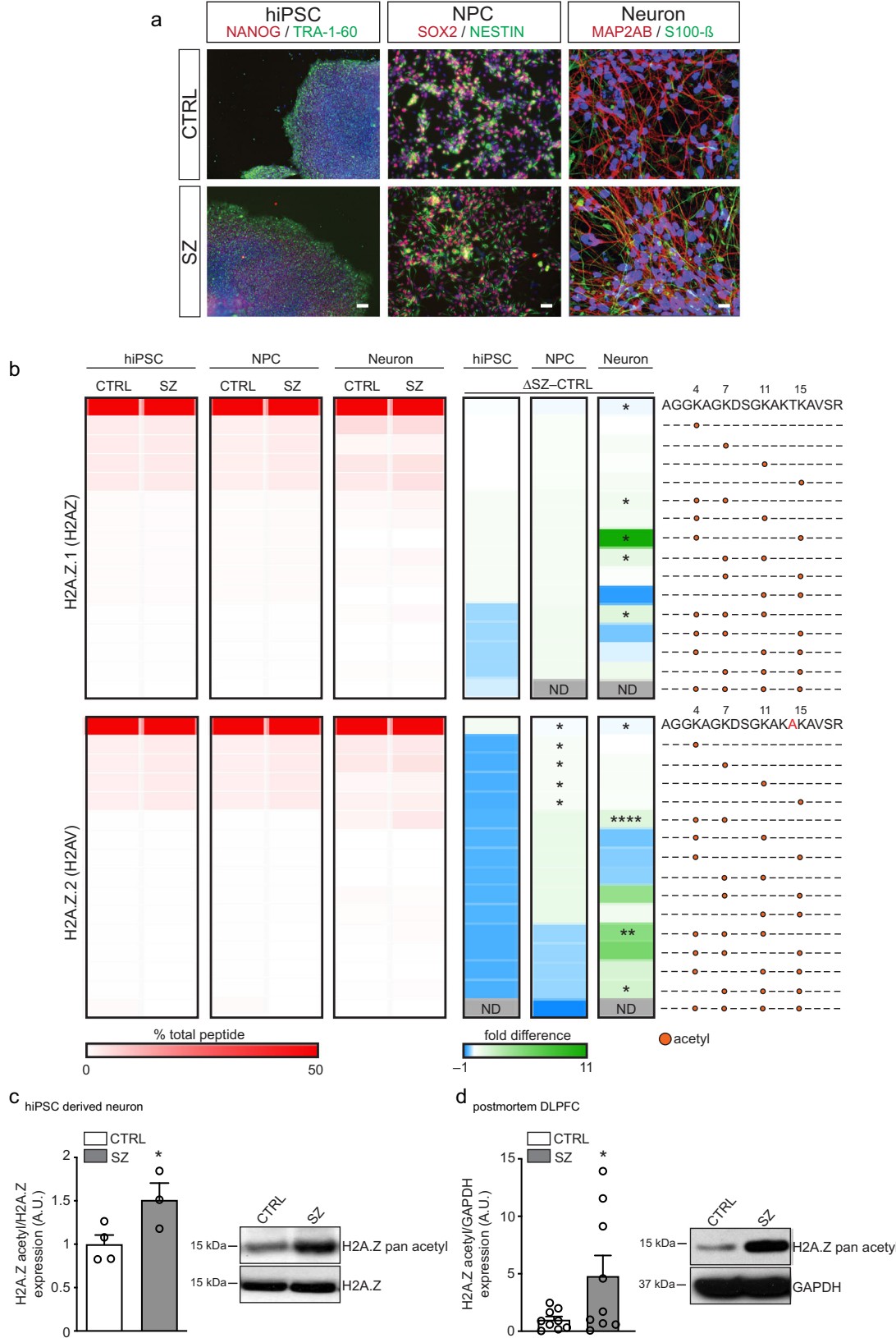

helps to explain the ITC results, demonstrating that Bromo1 has weaker binding affinity to H2A.Z peptides in comparison to Bromo2 (Fig. 2f, iv). To further validate this key residue involved in binding, we carried out a mutagenesis-based study. As shown in Fig. 2g, a

N433A mutant of BRD4$_{Bromo2}$ (which remains properly folded—see Supplementary Fig. 2a) completely loses its binding ability to H2A.Z (1–15) K4acK7acK11ac, indicating that K4ac recognition is essential for Bromo2-tri-acetylated H2A.Z binding.

**Fig. 1 Combinatorial histone hyperacetylation in SZ neurons. a** Representative (repeated >3X/group) patient-specific hiPSCs, NPCs, and neurons. Left-hiPSCs express NANOG (green) and TRA-1-60 (red). DAPI (blue). Center-hiPSC neural progenitor cells (NPCs) express NESTIN (green) and SOX2 (red). DAPI (blue). Right-hiPSC neurons express S100 calcium-binding protein B (green) and the dendritic marker MAP2AB (red). DAPI (blue). Scale bar 100 μm. **b** Heatmap depicting LC-MS/MS data for relative enrichment values of (un)modified and acetylated histones H2AZ.1 and H2AZ.2 in hiPSCs ($n = 3$/group), NPC ($n = 3$/group), and 4-week-old neurons from SZ vs. matched controls ($n = 4$/group). Absolute values (% total peptide) for each peptide are provided. Fold differences between SZ vs. controls are represented (biologically independent replicates/cell-type/condition). Heatmap data represented as means, *$p \leq 0.05$, **$p \leq 0.01$, ****$p \leq 0.0001$ (two-tail Student's $t$-tests performed within cell-type, SZ vs. CTRL; adjustments were not made for multiple comparisons). Please see Supplementary Data 1 for LC-MS/MS source data. Increased patterns of H2A.Z acetylation were confirmed via western blotting in **c** 4-week-old hiPSC neurons [$n = 3$ (SZ) vs. 4 (CTRL) biologically independent replicates], *$p = 0.0538$ (two-tail Student's $t$-tests), and in **d** DLPFC from biologically independent postmortem SZ subjects vs. matched controls ($n = 9$ per group; two-tail Student's $t$-tests), *$p = 0.0510$. Total H2A.Z and GAPDH were used as normalization controls, respectively. A.U. = arbitrary units (normalized to CTRL samples). Data are presented as averages ± SEM. Source data are provided in Source Data files.

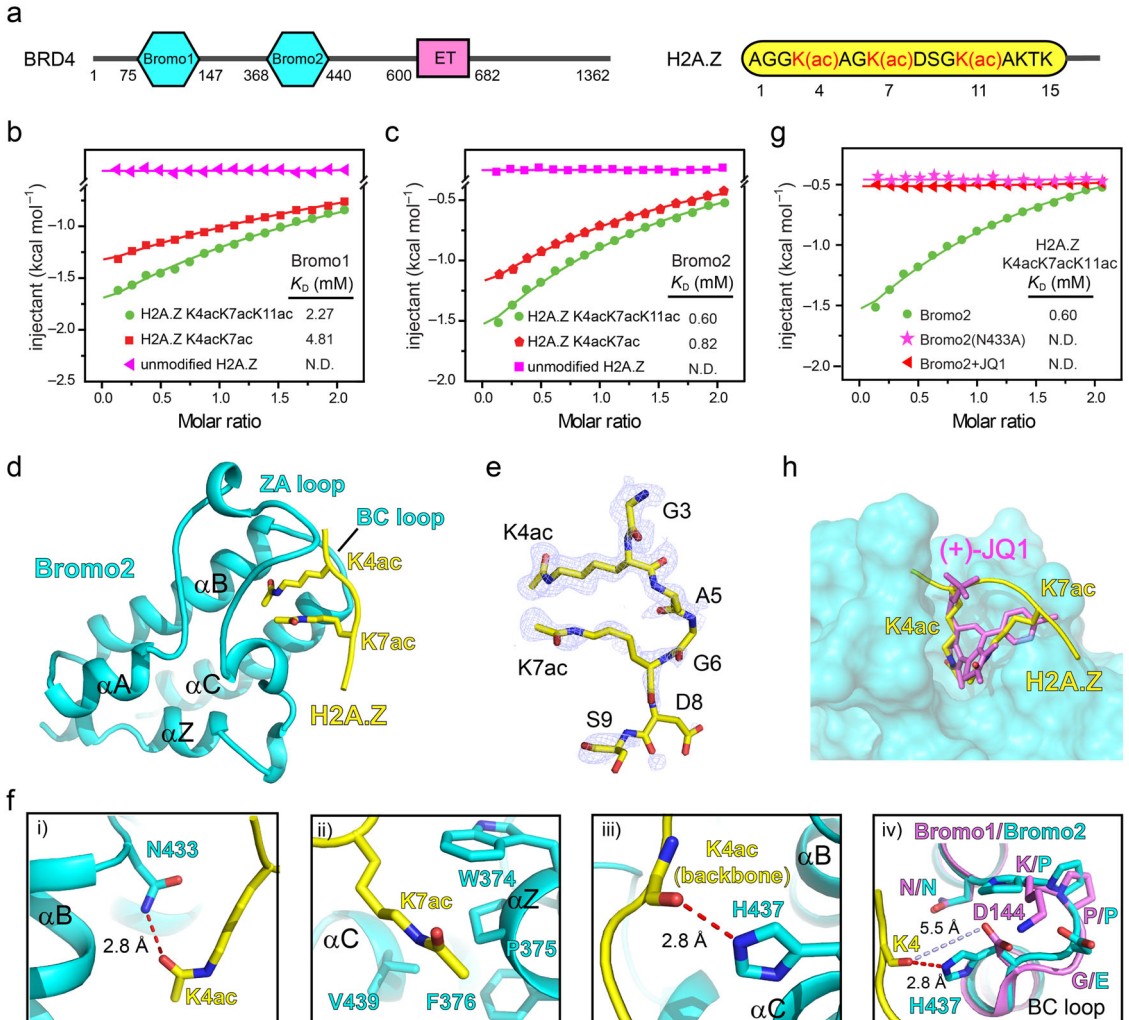

**Fig. 2 Molecular basis underlying BRD4 and H2A.Zac interactions. a** Domain architecture of BRD4 and H2A.Z peptide sequence. ITC fitting curves for **b** BRD4$_{Bromo1}$ and **c** BRD4$_{Bromo2}$ titrated with unmodified, di- and tri-acetylated H2A.Z (1–15) peptides. **d** The overall structure of BRD4$_{Bromo2}$ in complex with the H2A.Z (1–12) K4acK7acK11ac peptide. Bromo2 (cyans) and the histone H2A.Z peptide (yellow) are shown as ribbons; K4ac and K7ac of H2A.Z (yellow) are depicted as sticks. **e** Fo-Fc map of H2A.Z peptide in H2A.Z$_{Bromo2}$ complex counters at 2.0 σ and is shown as light-blue meshes. **f** Interaction details of H2A.Z (1–12) K4acK7acK11ac peptide recognized by BRD4$_{Bromo2}$, and structural alignment of Bromo1 and 2. K4ac (i), K7ac (ii), K4 (iii), and K4 (iv) of H2A.Z are shown as yellow sticks. Hydrogen bonds and distance are shown as red and light-blue dashes, respectively. Key residues of Bromo1 and Bromo2 are depicted as cyans and violet sticks, respectively. **g** ITC fitting curves of H2A.Z (1–15) K4acK7acK11ac peptide titrated into wild type, mutant and (+)-JQ-1 saturated Bromo2 of BRD4. N.D. = not detected. **h** Structural alignment of (+)-JQ-1 (violet) and H2A.Z peptide (yellow) in complex with BRD4$_{Bromo2}$ (cyans surface).

While our data certainly indicated that BRD4$_{Bromo2}$ can bind to polyacetylated H2A.Z, we also sought to investigate whether multivalent domain interactions may further prompt such binding. To do so, we performed ITC using the Bromo1-linker(GGS)$_5$-Bromo2 of BRD4 titrated with H2A.ZK4ac7ac11ac peptides cross-linked by a C-terminal cysteine (Supplementary Fig. 2b). Compared with Bromo2 and Bromo1 of BRD4, the extended Bromo1-linker-Bromo2 of BRD4 promoted further binding of BRD4 towards polyacetylated H2A.Z by 1.7- and 6.49-fold, respectively.

To test whether a well-known and clinically relevant inhibitor for BRD4 (and other BET family proteins), JQ1[18], may similarly disrupt interactions between BRD4 and hyperacetylated H2A.Z, we next explored its impact on tri-acetylated H2A.Z recognition by BRD4$_{Bromo2}$. As demonstrated in Fig. 2g, JQ1 abolishes binding of BRD4$_{Bromo2}$ to tri-acetylated H2A.Z, and structural analysis based upon Bromo2-(+)-JQ1 and Bromo2-tri-acetylated H2A.Z complex alignments indicate that JQ1 appropriately occupies the binding pocket of BRD4$_{Bromo2}$ for tri-acetylated H2A.Z (Fig. 2h). Additionally, according to sequence alignments of BET family proteins, the acetyl binding pocket is relatively conserved. Therefore, we performed additional ITC assessments of other BET family members against H2A.Zac vs. unmodified peptides. As expected, BRD2$_{Bromo1/2}$ and BRD3$_{Brono1/2}$ can also bind to polyacetylated H2A.Z (Supplementary Fig. 2c, d), suggesting that inhibition of multiple BET proteins simultaneously (as in the case of JQ1) may prove beneficial in the amelioration of SZ-related gene expression deficits.

Finally, to verify that BRD4 and H2A.Zac indeed overlap genome-wide in human cells, we performed chromatin immunoprecipitation (ChIP)-seq for pan acetylated H2A.Z in HeLa cells and correlated its enrichment with BRD4 occupancy at genic promoters. BRD4 and H2A.Zac genomic enrichments were found to highly correlate (Spearman's rank correlation = 0.88, $p < 2e$-16; Supplementary Fig. 3a), displayed nearly identical patterns of enrichment at target loci (e.g., the *Fos* locus; Supplementary Fig. 3b) and co-enriched genes targets were found to be transcriptionally sensitive to BET family inhibition via JQ1 treatments (Supplementary Fig. 3c). In all, these data demonstrate that BRD4 (likely along with related BET family proteins) is a bona fide "reader" of combinatorially acetylated H2A.Z and suggest that BET family inhibition may, at least in part, reverse specific disease related deficits linked to aberrant histone acetylation in SZ.

**BET family protein inhibition ameliorates SZ-related gene expression changes.** In an effort to explore potential effects of BET family protein inhibition on SZ, we performed RNA-seq on 4-week-old hiPSC-derived neurons from SZ cases and controls, as well as JQ1-treated cells (24 h, 250 nM JQ1 or dimethyl sulfoxide (DMSO)-vehicle control) (Supplementary Fig. 4a, b). Competitive gene-set enrichment analysis across a collection of 698 manually curated neural gene sets (subdivided into 8 categories), found strong enrichment of pre- and postsynaptic gene sets following JQ1 treatment, while SZ lines were correlated with presynaptic, brain-related and neuropsychiatric disorder gene sets (Fig. 3a). When modeling the additive effect of differential expression in SZ and JQ1-treated control samples computationally[19,20], we found that only enrichment in disorder gene sets remained, which was largely confirmed in SZ neurons treated with JQ1 (Fig. 3a).

To examine this interaction of SZ and JQ1 treatment in more detail, we performed two types of analyses: weighted gene co-expression network analysis (WGCNA, Supplementary Fig. 5a) and modeling of synergistic effects between SZ affliction and JQ1 treatment (Fig. 4). WGCNA resulted in 19 gene modules, of

which three (darkturquoise, salmon and darkred) were highly correlated with the SZ sample signature ($p = 0.04$, 0.04, 0.009, respectively) (Fig. 3b). In contrast, these modules were not correlated with the transcriptional profile of JQ1-treated SZ samples ($p = 0.9$, 0.4, 0.1, respectively), suggesting their SZ-related expression changes were ameliorated by inhibition of BET proteins. Over-representation analysis in our curated gene sets demonstrated their enrichment in a multitude of SZ and other neuropsychiatric gene sets (Fig. 3c).

While treatment of SZ neurons with JQ1 appeared to match an additive effect model, as confirmed by hierarchical clustering of the t-statistics for each contrast (Figs. 3a and 4a), we were interested in additional synergistic effects, particularly since a number of SZ regulated transcripts (as demonstrated via qPCRs) appeared to be regulated synergistically in the context of JQ1 treatments (Supplementary Fig. 5b). Genes that were significantly differentially expressed in comparison to the additive model were grouped into synergism categories. Genes were classified as "more" differentially expressed in the JQ1 treatment of SZ than predicted, if their logarithmic fold-change differed by at least the average standard error in all samples (Fig. 4c). Most genes were altered approximately as predicted or less, while 3.8% (673 genes) were more downregulated and 2.5% (441 genes) more upregulated than expected (Fig. 4c). Over-representation analysis revealed that genes more downregulated than simulated in the additive model were significantly enriched for SZ GWAS genes and abnormal head morphology (Fig. 4d, e). In summary, JQ1-dependent transcriptional changes led to partial amelioration of SZ-related gene expression abnormalities.

## Discussion

Here, we performed histone PTM LC-MS/MS analyses on hiPSCs, NPCs, and neurons derived from SZ cases vs. matched controls, and identified aberrant patterns of combinatorial hyperacetylation on histones H2A.Z and H4 in SZ neurons. Furthermore, we found that the bromodomain containing protein, BRD4—already known to interact with H4 acetylation signatures[16]—is also a bona fide "reader" of combinatorial H2A.Z acetylation, with pharmacological inhibition of these BET family proteins resulting in alleviation of SZ-related gene expression. Thus, our data suggest that therapeutic strategies aimed at reversing deleterious interactions between BET proteins and hyperacetylated histone PTM signatures may prove beneficial in preventing or treating certain aspects of the disorder.

Histone variant proteins, such as H2A.Z, are expressed in brain in a replication-independent manner vs. their canonical counterparts, which require active cell cycling for chromatin incorporation[16]. H2A.Z exists as two isoforms that differ by only three amino acids (H2A.Z.1, H2A.Z.2), however, it remains incompletely understood as to exactly how H2A.Z may regulate transcriptional processes in post-replicative cells. Depending upon its modification state, H2A.Z has been shown to associate with both transcriptional repression (e.g., monoubiquitination by the PRC1 complex[21]) and activation (acetylation)[22–24], yet only a handful of studies have begun to identify definitive links between H2A.Z variant proteins and neurological processes/disorders. For example, knockdown of H2A.Z.1 in primary neurons results in altered expression of candidate genes linked to autism and SZ[25], and conditional knockout mice display abnormal dendrites during development[26]. Knockout animals also exhibit behavioral deficits, including impairments in learning and memory, which are reminiscent of cognitive deficits associated with SZ.

BET family proteins, which include BRD2, BRD3 and BRD4, are chromatin associated "reader" proteins that interact with acetylated lysine residues on histone proteins to assemble

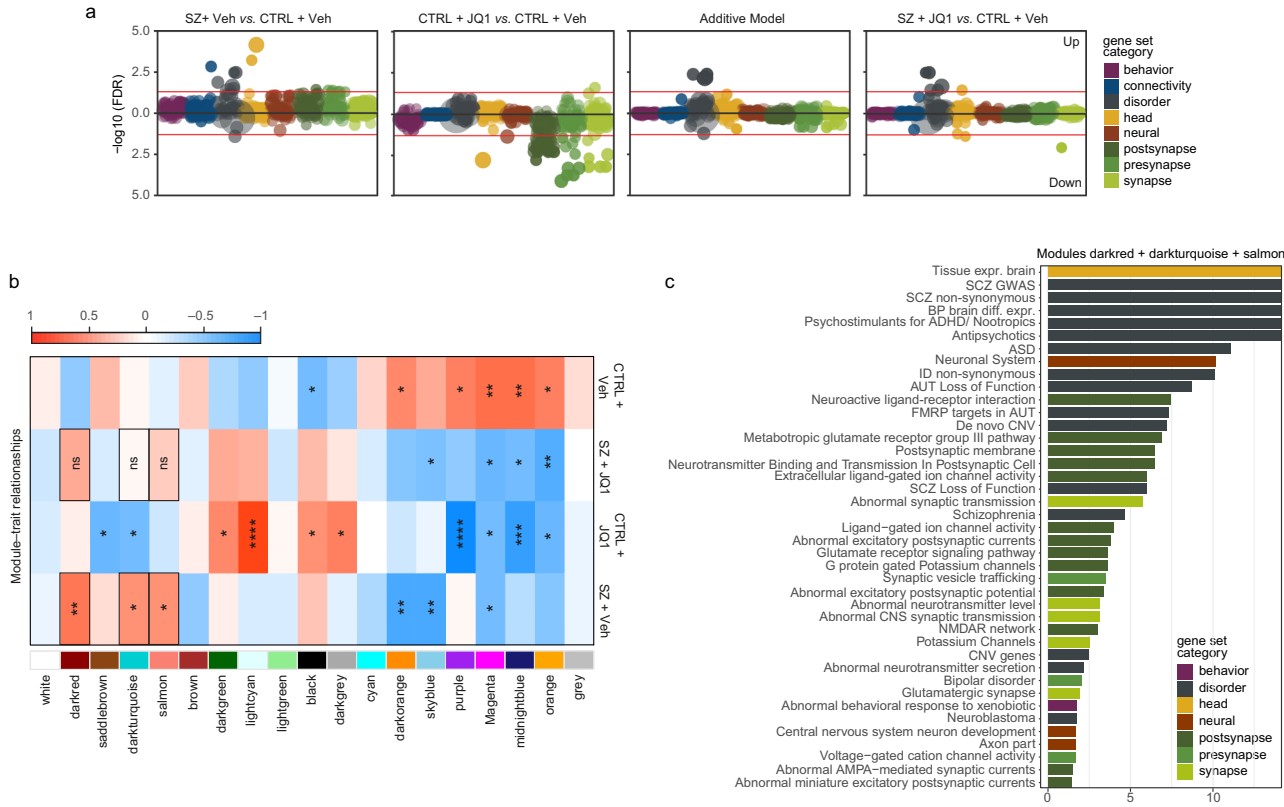

**Fig. 3 RNA-seq of SZ, control (CTRL) and JQ1-treated hiPSC neurons. a** Competitive gene-set enrichment analysis (camera) of differential expression contrasts from the current study, based on 698 curated neural gene sets[19, 20], stratified by eight categories. **b** Sample trait—module association heatmap. Columns correspond to module eigengenes, rows to sample groups. Cell labels denote weighted Pearson correlation and Student asymptotic *p*-value of corresponding module and group ($n = 4$ ctrl + vehicle samples, $n = 3$ SZ + vehicle samples, $n = 4$ Ctrl + JQ samples and $n = 3$ SZ + JQ samples). *$p \leq 0.05$, **$p \leq 0.01$, ****$p \leq 0.0001$. **c** Over-representation analysis (ORA), using a hypergeometric test, of 698 curated gene sets and ranked genes in three modules, Darturquoise, Salmon, and Darkred. These showed highest association with the SZ samples' transcriptional signature and reduced correlation with SZ + JQ samples. Only gene sets with enrichment FDR < 5% are shown.

chromatin complexes and recruit transcriptional activators to specific promoters[27,28]. While it has been shown that the combination of H2A.Z and H4 acetylation at target loci results in the potentiation of BET family protein recruitment (e.g., BRD2)[29], direct structural evidence for acetylated H2A.Z interactions with BET proteins has been lacking. Here, we established, via both ITC and x-ray crystallography assessments, that BRD4 is a bona fide "reader" of acetylated H2A.Z. Given that BRD4-acetylation interactions are prominent during early stages of cellular proliferation and differentiation[30,31], and considering that histone hyperacetylation was identified prominently in maturing neurons derived from SZ patients, this hints at the possibility that neurodevelopmental components of SZ, particularly those occurring during early stages of neuronal differentiation, may be potentiated by aberrant states of H2A.Z acetylation.

Beyond the identification of aberrant roles for bromodomain-acetylation interactions in the potentiation of various cancers[32], this family of proteins has also gained recent attention for its potential roles in promoting specific psychiatric disorders. For example, interaction networks for BRD1 have demonstrated enrichment for SZ risk genes and have provided evidence of enhanced binding to gene promoters associated with brain development and susceptibility to mental disorders[33]. Additionally, recent rodent studies have demonstrated an upregulation of *BRD1* gene expression following periods of chronic stress[34], and mice heterozygous for a targeted deletion of *BRD1* display behavioral phenotypes with broad translational relevance to psychiatric disorders[35,36].

BET inhibitors, such as JQ1, function by competitively binding to acetylated histone lysine residues to prevent interactions between BET proteins and chromatin. Although these drugs have gained notable attention as anti-tumor agents in the treatment of certain cancers[37–40], as well as with respect to their abilities to attenuate the rewarding aspects of cocaine exposures[41], the potential of using such inhibitors to alleviate SZ-related gene expression has remained unexplored. We provide evidence suggesting that treatments with the BET family inhibitor, JQ1, can partially rescue transcriptional deficits associated with disease. Our work suggests that BET inhibitors may serve as additional, promising therapeutics for modulating gene expression aberrations associated with SZ and warrant further investigations into precisely how these mechanisms may mediate SZ susceptibility during brain development.

## Methods
**hiPSC and NPC culture.** Fibroblasts from four cases (three males and one female of Caucasian/Jewish/Scandinavian decent—two Proband, one Proband 1, and one Sibling 1) and four controls (two Caucasian males and two Caucasian females) were obtained from the Coriell Cell Repository (Camden, NJ, USA) or American Type Culture Collection (Manassas, VA, USA) and reprogrammed as described previously using tetracycline-inducible lentiviruses expressing *OCT4, SOX2, KLF4, cMYC,* and *LIN28*, driven by the reverse tetracycline transactivator (rtTA)[14]. Following 2 weeks of embryoid body differentiation in N2 medium, visible rosettes were manually dissected onto poly-ornithine/laminin-coated plates. Rosettes were cultured in NPC medium (Dulbecco's modified Eagle medium (DMEM)/F12, 1x N2, 1x B27-RA (Life Technologies), 1 µg ml⁻¹ Laminin and 20 ng ml⁻¹ FGF2) and dissociated in TrypLE (Life Technologies) for 3 min at 37 °C. NPCs were maintained at high density, grown on

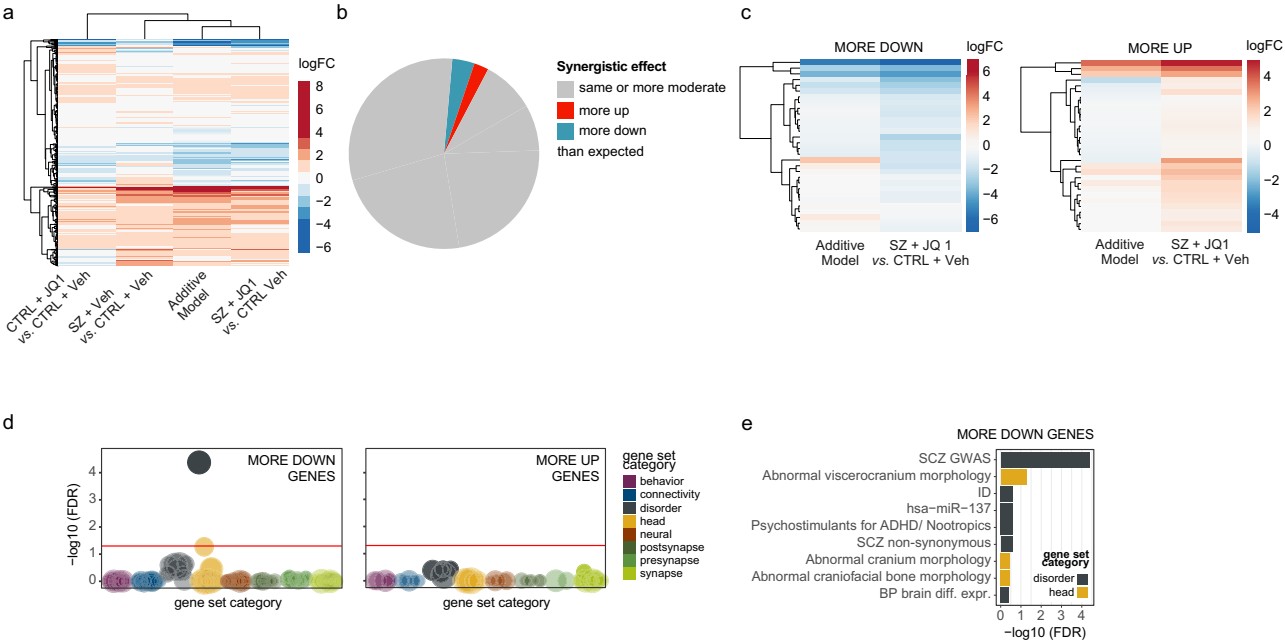

**Fig. 4 Synergistic effects of SZ and JQ1 treatment in hiPSC-derived neurons. a** Hierarchical clustering of the *t*-statistics for all comparisons. The color gradient represents *t*-statistic values. **b** Pie chart showing percentages of genes that exhibit similar or more moderate differential expression (gray) in SCZ samples following JQ treatment, in comparison with the additive model, as well as genes that are more downregulated (blue) and more upregulated (red). **c** Hierarchical clustering of the differential expression log2 fold-changes of "more downregulated" and "more upregulated" genes, in the additive model vs. the JQ1-treated SZ neurons. **d** Over-representation analysis (ORA) using a hypergeometric test, of 698 curated gene sets and those "more down" and "more up" genes with synergistic differential expression, ranked by adjusted significance. **e** Bar chart showing more detailed ORA results of the "more down" genes as in **d**.

poly-ornithine/laminin or Matrigel (BD)-coated plates in NPC medium and split ~1:4 every week with Accutase (Life Technologies)[14,42]. For neuronal differentiations, NPCs were dissociated with Accutase and plated in neural differentiation medium (DMEM/F12, 1× N2, 1× B27-RA, 20 ng ml$^{-1}$ BDNF (Peprotech), 20 ng ml$^{-1}$ GDNF (Peprotech), 1 mM dibutyryl-cyclic AMP (Sigma), 200 nM ascorbic acid (Sigma) onto poly-ornithine/laminin-coated plates and matured for 4-weeks. For drug experiments, cells were treated with 250 nM JQ1, or DMSO vehicle control, for 24 h prior to harvest.

Since all research described in this manuscript was performed on deidentified human samples obtained for broadly consented scientific research by either American Type Culture Collection (ATCC) or the Coriell Cell Repository, this work was found to be exempt by the Internal Review Committee of the Icahn School of Medicine at Mount Sinai. This work was also reviewed by the Embryonic Stem Cell Research Oversight Committee at the Icahn School of Medicine at Mount Sinai, and all work was conducted in accordance to the criteria set by the Declaration of Helsinki.

**HeLa cell culture**. HeLa cell lines were obtained from the American Type Culture Collection (ATCC) and grown in DMEM supplemented with 10% fetal bovine serum and 100 μg ml$^{-1}$ streptomycin/100 U ml$^{-1}$ penicillin. Cells were maintained at 37 °C in a 5% $CO_2$, 95% humidified incubator. For drug experiments, cells were treated with 500 nM JQ1, or DMSO vehicle control, for 24 h prior to harvest.

**Human postmortem tissues**. Human brain tissues from the UT Neuropsychiatry Research Program (Dallas Brain Collection-DBC) were collected from the Southwestern Institute of Forensic Sciences at Dallas, UT Southwestern Transplant Services Center, and UT Southwestern Willed Body Program; after consent from donor subjects' next of kin was received, along with permission to access medical records and to hold a direct telephone interview with a primary caregiver. All clinical and medical information obtained for each donor is reviewed by three research psychiatrists, using DSM-V criteria for diagnoses. Blood toxicology screens for drugs of abuse, alcohol and prescription drugs, including psychotropics, are conducted for each donor subject from the Southwestern Institute of Forensic Sciences at Dallas. Collection of postmortem human brain tissues is approved by the University of Texas Southwestern Medical Center Institutional Review Board [STU 102010-053]. During brain dissections, tissue samples are removed, frozen immediately using dry ice and 2-methylbutane (1:1, v:v) and stored at −80 °C. For all western blotting validation experiments, GAPDH was used an internal reference control for the best normalization and most reliable indicator of equal protein concentration. Demographic information can be found in Supplementary Table 1.

**Immunocytochemistry**. Cells were fixed in 4% paraformaldehyde in PBS at 4 °C for 10 min, permeabilized at room temperature for 15 min in 1.0% Triton in PBS and blocked in 5% donkey serum with 0.1% Triton at room temperature for 30 min. The following primary antibodies and dilutions were used: goat anti-Nanog (R&D, AF1997), 1:200; mouse anti-Tra1-60 (Millipore, MAB4360), 1:100; mouse anti-human Nestin (Millipore, ABD69), goat anti-Sox2 (Santa Cruz, sc-17320), 1:200; rabbit anti-βIII-tubulin (Covance, PRB-435P), 1:200; mouse anti-MAP2AB (Sigma, M1406), 1:200; mouse anti-S100b (Sigma-Aldrich, S2532), 1:1000. Secondary antibodies were Alexa donkey anti-rabbit 488 (Jackson Immuno 711-545-152) and 568 (Life Technologies A10042), Alexa donkey anti-mouse 488 (Jackson Immuno 715-545-151) and 568 anti-mouse (Life Technologies A10037), and Alexa donkey anti-goat 488 (Jackson Immuno 705-545-147) and 568 (Jackson Immuno 705-605-147); all were used at 1:300. To visualize nuclei, slides were stained with 0.5 μg ml$^{-1}$ DAPI (4′,6-diamidino-2-phenylindole) and then mounted with Vectashield.

**Histone preparation and mass spectrometry analysis**. Histones were acid extracted from nuclei and ~10 μg of material per sample was chemically derivatized, followed by digestion to tryptic peptides[43]. Prior to LC-MS analysis, derivatized samples were desalted using C18 stage-tips. Using a 75 μm ID x 17 cm, in-house packed (with Reprosil-Pur C18-AQ, 3 μm; Dr. Maisch GmbH, Germany) nano-column fitted onto an EASY-nLC nanoHPLC (Thermo Scientific, San Jose, Ca, USA), peptides were separated on an HPLC gradient comprising 2% to 28% solvent B ($A = 0.1\%$ formic acid; $B = 95\%$ MeCN, 0.1% formic acid) over 45 min, from 28% to 85% solvent B in 5 min and 85% B for 10 min at a flow-rate of 300 nl min$^{-1}$. This nLC was coupled online to an LTQ-Orbitrap Elite mass spectrometer (Thermo Scientific), and data were acquired using data-independent acquisition (DIA). Briefly, full scan MS ($m/z\ 300 - 1100$) was acquired in the Orbitrap with a resolution of 120,000 (at 200 $m/z$) and an AGC target of 5x10e5. MS/MS was done in centroid mode in the ion trap with sequential isolation windows of 50 $m/z$ with an AGC target of 3x10e4, a CID collision energy of 35 and a maximum injection time of 50 ms. Data were analyzed using the in-house software, EpiProfile[44], wherein peptide relative ratios were calculated using the total area under the extracted ion chromatograms of all peptides with the same amino acid sequence (including all of their modified forms) as 100%. LC-MS/MS quantifications are summarized in Supplementary Data 1.

**Western blotting**. Proteins were electrophoresed on 18% or 4–12% NuPAGE™ Bis-Tris Protein Gels (Invitrogen) and transferred to nitrocellulose membranes. Protein transfer was always confirmed with direct blue staining (0.1% stock aqueous solution in MilliQ water) before being incubated with primary antibodies

(listed below) overnight at 4 °C (all membranes were blocked in 5% milk). Membranes were then washed and incubated with peroxidase-labeled secondary antibodies (1:2000–1:10,000 depending on the primary antibody used). Bands were visualized using Immobilon Western Chemiluminescent HRP Substrate (Millipore). Where appropriate, bands were quantified with NIH Image J Software and bands were normalized to their appropriate control for equal loading. To best control for postmortem brain tissue variability, GAPDH or H4 were used as the loading controls for normalization due to their stable and reliable expression across cases and controls (note that total H2A.Z antibodies did not yield high-quality data in postmortem western blots, which is why GAPDH was used as a normalization control for H2A.Zac in those experiments).

**Antibodies used**. All primary antibodies used in this study were validated, where appropriate, for use in immunoblotting, immunocytochemistry (ICC)/immuno-histochemistry (IHC), and/or ChIP-seq experiments.

*Western blotting.* Histone H2AZ (Active Motif; Cat#39113; Lot#06217001; 1:1000), Histone H2AZ.pan Acetyl (Active Motif; Cat#39642; Lot#26409001; 1:500), Histone H4 (Abcam; Cat#Ab10158; Lot#GR3268080-1; 1:10,000), Histone H4K5K8K12acetyl (GeneTex; Cat#GTX60337; Lot#821900948; 1:500), GAPDH (Sigma Cat#G9545; Lot#015M4824V; 1:1000).

*ChIP-seq.* Histone H2AZ.pan Acetyl (GeneTex; Cat#GTX60813; Lot#822002999; 7.5 μg/IP).

**Plasmids and peptides**. The cDNAs encoding the BRD4-bromo1 (amino acids 54–168) and BRD4-bromo2 domain (amino acids 351–457) were cloned into pSUMOH10 vectors (modified based on pET28b) and pGEX6p vectors (GE Healthcare), respectively. Point mutations were generated using the QuikChange site-directed mutagenesis kit (Stratagene) following the manufacturer's instructions. BRD4$_{bromo1}$ and BRD4$_{bromo2}$ was fused by a $(GGS)_5$ linker and then cloned into pGEX6p vectors (GE Healthcare). The cDNAs encoding the BRD2$_{bromo1}$ (amino acids 73–194), BRD2$_{bromo2}$ domain (amino acids 455–347), BRD3$_{bromo1}$ (amino acids 24–144) and BRD3$_{bromo2}$ domain (amino acids 306–416) were cloned into pGEX6p vectors (GE Healthcare). Histone peptides bearing different modifications were synthesized at SciLight Biotechnology.

**Protein expression and purification**. The bromo1 and bromo2 domains of BRD4 were expressed with an N-terminal HIS 10-SUMO tag and N-terminal GST-tag, respectively, in *Escherichia coli* strain BL21 (DE3) in the presence of 0.2 mM iso-propyl β-D-thiogalactoside (IPTG) at 18 °C in LB medium. After overnight induction, cells were harvested by centrifugation and re-suspended in lysis buffer: 500 mM NaCl, 20 mM Tris-HCl, pH 7.5, and then disrupted by an EmulsiFlex-C3 high-pressure homogenizer (Avestin). Cell lysates were subjected to centrifugation at $16,770 \times g$, and the supernatants were applied to a GST affinity column or His affinity column. Proteins were directly digested on the column overnight by the PreScission proteases or SUMO protease ULP1. The resultant proteins were further purified by anion-exchange chromatography using HiTrap Q columns (GE Healthcare), followed by use of size exclusion chromatography Superdex G75 columns (GE Healthcare) with the elution buffer containing 20 mM Hepes-Na, pH 7.5, 100 mM NaCl, and 5 mM DTT. The peak fractions were pooled and concentrated to ~9 mg ml$^{-1}$, aliquoted and stored at −80 °C for future use.

Mutant BRD4-Bromo2 N433A, BRD2$_{bromos1/2}$, BRD3$_{bromos1/2}$, and Bromo1-linker(GGS)$_5$-Bromo2 of BRD4 were purified using the same procedure as described above.

**Crystallization, data collection, and structural determination**. Crystallization was performed by the sitting-drop vapor diffusion method under 18 °C by mixing 1 μl of protein with 1 μl of reservoir solution. Crystals of free BRD4-brmo2 were grown from the solutions containing 0.1 M Amino acids, 0.1 M Buffer System 3, pH 8.5, 50% v/v Precipitant Mix 4 (morpheus). Then the H2A.Z (1–20) K4acK7acK11ac peptides were soaked into the crystals for 1 day at a molar ratio of 1:40.

For data collection, the crystals were flash frozen (100 K) in the above reservoir solution. A diffraction dataset was collected to 1.5 Å resolution at wavelength 0.9793 Å on beamline BL17U at the Shanghai Synchrotron Radiation Facility. All data were indexed, integrated and merged using the HKL2000 software package.

The structure of BRD4-Bromo2/H2A.ZK4acK7acK11ac was solved by molecular replacement using MOLREP35 with the free BRD4-Bromo2 structure as a search model. The structure was refined using PHENIX, with iterative manual model building using COOT. Detailed structural refinement statistics are shown in Supplementary Information Table S4. All structural figures were created using PYMOL.

**Isothermal titration calorimetry (ITC) measurements**. All of the calorimetric experiments for wild type or mutant proteins were conducted at 25 °C with a MicroCal PEAQ-ITC instrument in 20 mM Hepes-Na, pH 7.5, and 100 mM NaCl buffer. In all, 3 mM of H2A.Z peptides were titrated into 0.3 mM of

bromodomains. Owing to low solubility of (+)-JQ1(MCE) in water, 0.03 mM (+)-JQ1 and 0.03 mM BRD4-Bromo2 were mixed overnight to obtain the (+)-JQ-1 prebonded protein. The acquired calorimetric titration curves were analyzed with Origin version 7.0 software (OriginLab) using the "One Set of Binding Sites" fitting model.

**Circular dichroism (CD)**. Circular dichroism measurements were performed with a Chirascan plus CD spectrometer. Data were collected with 0.1 mg ml$^{-1}$ protein in 10 mM NaCl, 2 mM Hepes (pH 7.5) buffer over a wavelength range of 190-260 nm, with 1 nm increments, in a 0.5 mm pathlength rectangular cuvette at 25 °C. All measurements were performed in triplicate.

**ChIPs, ChIP-seq library preparations, and analyses**. H2A.Zac ChIPs and ChIP-seq library preparations from HeLa cells were carried out as previously described[45] and according to Illumina protocols. They were then sequenced with an Illumina HiSeq4000 Sequencing_High Output mode v4.

Raw sequencing reads were mapped to hg38 using HISAT2[46]. Uniquely mapped reads were retained. Alignments were filtered using SAMtools to remove duplicate reads, and peak-calling—normalized to respective inputs—was performed using MACSv2.1.1[47] with default settings and filtered for $p < 0.05$ and fold-change > 1.2. Peaks called were annotated using region_analysis software and saved in a tab delimited text file.

*Corrgram.* BRD4 DMSO samples were obtained from GSE151038. TSS sites for all genes in the genome were found using Ensembl BioMart. Promoter regions were defined as +/− 1 kb from the TSS sites. Counts of reads mapping to the promoter regions for H2A.Zac and BRD4 samples were obtained using featureCounts. Raw counts were converted to RPKM and the corrgram R package[48] was utilized to compute Spearman correlation between the coverage of promoter regions for BRD4 and H2A.Zac datasets.

The NGS-Data-Charmer repository, which hosts an automated NGS data analysis pipeline, was used in ChIP-seq analyses (https://github.com/shenlab-sinai/NGS-Data-Charmer).

**RNA extraction, qPCRs, and library preparation**. Cell pellets from vehicle (PBS) and JQ1 treated and neurons were homogenized in Trizol (Thermo Fisher; cat#15596026) and processed according to the manufacturer's instructions. RNeasy Microcolumns (Qiagen; cat#74004) were used to further purify RNA, and a nanodrop spectrophotometer confirmed RNA 260/280 and 260/230 ratios to be >1.8. Following RNA purifications, samples were either (1) reverse transcribed to cDNA and assessed via qPCR using SYBR Green (see Supplementary Table 3 for qPCR primers), or (2) prepared into RNA-seq libraries according to Illumina protocols and sequenced with an Illumina HiSeq4000 Sequencing_High Output mode v4.

**RNA sequencing**. All RNA-Seq analyses were performed in R (www.R-project.org). All heat maps and their hierarchical clustering were computed using the pheatmap package. If the number of values to be clustered exceeded 1000, rows were pre-aggregated using *k*-means clustering.

**Data preprocessing**. Genes with over 15 counts in at least 2 samples were retained for further analysis. Normalization factors for library sizes were calculated using the trimmed mean of *M*-values (TMM) method in the calcNormFactors function (edgeR package). For the purpose of linear modeling, "JQ treatment" and "Diagnosis" were combined into one group variable.

**Differential gene expression analysis**. The limma voom function was used to compute weights for heteroscedasticity adjustment by estimating the mean-variance trend for log2 counts. Linear models were fit to the expression values of each gene using the lmFit function: Gene expression ~ group. The coefficients, standard deviations and correlation matrix were then recalculated, using contrasts.fit in terms of the comparisons of interest. Empirical Bayes moderation was applied using the eBayes function to obtain more precise estimates of gene-wise variability. *p*-values were adjusted for multiple hypotheses testing using false-discovery rate (FDR) estimation, and differentially expressed genes were determined as those with an estimated FDR ≤ 5%, unless stated otherwise.

**Concordance analysis**. Concordance between differential expression in the tested comparisons and one external dataset (postmortem (CMC, common mind consortium) was evaluated through Spearman correlation of t-statistics.

**Analysis of synergistic effects**. The expected additive effect was modeled through addition of the individual comparisons: (JQ vs. ctrl) + (SZ vs. ctrl)[20]. The synergistic effect was modeled by subtraction of the additive effect from the combinatorial comparison: (SZ + JQ vs. ctrl) – (JQ vs. ctrl)–(SZ vs. ctrl). Fitting of this model for differential expression gives genes that show a difference in the differential expression computed for an additive effect model and that computed for the

combinatorial experiment. However, interpretation of the resulting DEGs depends on several factors, such as the direction of fold-change in all three models. To identify genes of interest, namely those whose magnitude of change is larger in the combinatorial perturbation vs. the additive model, we categorized all genes by the direction of their change in both models and their log2FC in the synergistic model. First, log2FC standard errors (SE) were calculated for all samples. Genes were then grouped into "positive synergy" if their FC was larger than SE and "negative synergy" if smaller than -SE. If the corresponding additive model log2FC showed the same or no direction, the gene was classified as "more" differentially expressed in the combinatorial experiment than predicted. In all, 1115 genes were computed to be in this category (673 more down, 442 more up).

**Enrichment analysis.** Gene-set enrichment analysis (GSEA) was performed on a curated subset of the MAGMA collection (698 curated neural gene sets stratified by eight categories[20]) using the limma package camera function, which tests if genes are ranked highly in comparison to other genes in terms of differential expression, while accounting for inter-gene correlation. Owing to the small sample size in this study and moderate fold-changes overall, changes in gene expression may be small and distributed across many genes. However, similar to previous studies, more powerful enrichment analyses in the limma package were used. These evaluate enrichment based on genes that are not necessarily genome-wide significant and identify sets of genes for which the distribution of t-statistics differs from expectation.

**Over-representation analysis.** Over-representation analysis (ORA) was performed when subsets of DEGs were of interest, such as the synergistic "more up" and "more down" genes. The genes of interests were ranked by –log10 (p-value) and enrichment was performed against a background of all expressed genes using the WebGestaltR package.

**Weighted gene co-expression network analysis (WGCNA).** Unsigned co-expression networks were generated using the WGCNA R package from normalized and residualized expression data (17,792 genes) of all samples. An unsigned adjacency matrix was constructed using a soft thresholding power of 7 to maximize scale-free topology model fitting. The adjacency matrix was then transformed into a topological overlap matrix (TOM) to reduce noise, which in turn was transformed into a dissimilarity matrix by computing 1-TOM. Hierarchical clustering was performed on the TOM-based dissimilarity matrix. Finally, modules were identified using dynamic tree cut as a function of the hierarchical gene clustering and dissimilarity matrix. To merge modules with high similarity, module eigengenes (ME) were calculated, clustered as above and visually inspected for the minimum merging height cutoff, which was chosen at 0.17. The resulting 18 co-expression modules were then assigned a color and unconnected genes grouped into a "gray" module.

**General statistics.** For most statistical comparisons, Prism 9 [Version 9.3.1 (350)] for macOS Mojave (Version 10.14.6) was used. Two-tailed student's t-tests were used for all statistical comparisons comparing two groups to one another (all measurements were taken from distinct samples). All data involving statistics are presented as means ± SEM. In cases where only representative data are provided, experiments were repeated a minimum of three times.

**Reporting summary.** Further information on research design is available in the Nature Research Reporting Summary linked to this article.

## Data availability

The data that support this study are available from the corresponding authors upon reasonable request. No restrictions on data availability apply. The RNA-seq data generated in this study have been deposited in the National Center for Biotechnology Information Gene Expression Omnibus (GEO) database under accession number GSE144639. The atomic coordinates and structure factors have been deposited in the Protein Data Bank (PDB) under PDB ID code 6KO2. The histone modification mass spectrometry proteomics data have been deposited to the ProteomeXchange Consortium via the PRIDE partner repository with the dataset identified PXD031767. Source data are provided with this paper.

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

## Acknowledgements

We would like to thank members of the Maze, Brennand and Li laboratories for critical readings of the manuscript. We thank the staff members at beamline BL17U of the Shanghai Synchrotron Radiation Facility and the China National Center for Protein Sciences Beijing for providing facility support. This work was partially supported by grants from the National Institutes of Health: P50 MH096890 (I.M.), R01 HD097088 (I.M.), R01 MH116900 (I.M.), K99 MH120334 (L.A.F.), R01 AI118891 (B.A.G.), P01 CA196539 (B.A.G.), R56 MH101454 (K.J.B.), R01 MH106056 (K.J.B.) as well as awards from: (1) MQ Mental Health Research Charity, MQ15FIP100011 (I.M.), (2) Alfred P. Sloan Foundation Fellowship in Neuroscience (I.M.), (3) One Mind Institute (I.M.), (4) Brain and Behavior Research Foundation (L.A.F.), (5) the New York Stem Cell Foundation (K.J.B.) and (6) the National Natural Science Foundation of China (91753203 and 31725014, H.L.).

## Author contributions

I.M. and K.J.B. conceived of the project with input from L.A.F., S.Z., and H.L. L.A.F., S.Z., N.V.B., B.A.G., H.L., K.J.B., & I.M. designed the experiments and interpreted the data. L.A.F., S.Z., A.T., N.V.B., R.M.B., B.C., E.F., B.A.G., H.L., K.J.B., & I.M. collected and analyzed the data. N.S., A.R., J.C.C., & L.S. performed the sequencing-based bioinformatics with input from K.J.B. & I.M. K.G. & C.A.T. provided deidentified human postmortem brain tissues. L.A.F., S.Z., N.S., H.L., K.J.B., & I.M. wrote the manuscript.

## Competing interests

The authors declare no competing interests.

## Additional information

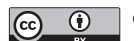

