## [Peer Review File · Nature Communications]

Reviewers' comments:

Reviewer #1 (Remarks to the Author):

In this report, Farrelly and colleagues characterize posttranslational modifications in human iPSC-derived neurons from individuals with schizophrenia. MS analysis reveals elevated acetylation level for several lysine residues of H2AZ in these neurons, and this finding was confirmed in postmortem human brain. The authors further show that bromodomains of BRD4 can bind polyacetylated H2AZ peptides albeit weakly, in a hundreds-micromolar to millimolar range, and the second bromodomain can be crystallized in complex with a H2AZK4ac/K7ac/K11ac peptide. Finally, the authors show that JQ1 treatment results in changes in gene expression in neurons and improves sensorimotor gating behavior in mice.

This work describes novel findings in an exciting area relevant to human health, though the manuscript could be further strengthened by incorporating additional more convincing data. The major weakness from my point of view is that the three parts although exciting are not as connected as they should, i.e. (i) finding that H2AZ is hyperacetylated at some sites (ii) BRD4 bromodomains bind acetylated H2AZ peptides however very weakly in comparison to binding of these domains to other acetylated, even monoacetylated histones. This obviously raises the question- is the weak association of BDs with H2AZ physiologically relevant? (iii) JQ treatment affects gene expression patterns which is expected granted JQ1 targets BRD4 BDs, but there is no evidence that H2AZ acetylation is involved here.

A few minor comments: binding affinity could be increased perhaps by testing linked bromodomains. Is N433A folded? A rationale to go after BRD4 could be strengthened by testing other known acetyllysine binding domains along with BRD4 BDs by pull-down experiments or peptide arrays.

Reviewer #2 (Remarks to the Author):

This manuscript identifies novel histone acetylation patterns of the H2A.Z variant histone proteins (e.g. H2A.Z.1/2-K4acK7acK11Ac) within human-induced pluripotent stem cell derived (hiPSC) forebrain neurons from individuals with schizophrenia (SZ) using mass spectrometry. Following this, the authors confirm that one of the bromodomains of the bromodomain containing protein BRD4 (BRD4Bromo2) has high affinity for, and interacts with, H2A.Z.K4acK7acK11ac using isothermal titration calorimetry. RNA-Sequencing analysis on hiPSC-derived neurons from SZ cases and controls that were treated with a BRD4 inhibitor, JQ1, reveals that BRD4 inhibition alters a subset of SZ-related gene expression changes. Lastly, chronic injections of JQ1 blocked amphetamine-induced hyperlocomotion in

mice, and increased prepulse inhibition in a different group of JQ1-treated mice. Overall, these studies demonstrate that BRD4 inhibition ameliorates transcriptional and behavioral changes typically seen in SZ cases and also that BRD4 binds acetylation sites that are hyperacetylated in hiPSCs-derived from SZ samples. These results significantly contribute to our understanding of how the epigenome is altered in SZ, how BRD4 may be interacting with hyperacetylated lysines, and the potential use of BRD4 inhibition to alter sensorimotor phenotypes.

Some additional experiments are suggested, but not necessary, prior to publishing this manuscript. Other minor issues mentioned below as well.

1) In vitro ITC studies demonstrate that 1) BRD4 interacts with H2A.ZK4acK7acK11Ac and 2) JQ1 abolishes binding of BRD4Bromo2 to tri-acetylated H2A.Z. In addition, RNA-sequencing studies find that application of the BRD4 inhibitor alters SZ-related gene expression in hiPSC-SZ samples. However, it may serve to benefit the conclusions made from this work by examining whether the mice that exhibited improvements in SZ-related behavioral impairments have decreased binding of BRD4 at H2A.ZK4acK7acK11Ac, or similar gene expression changes found from RNA-sequencing within the prefrontal cortex.

2) Behavioral studies were performed with chronic (7 day) treatment of the BRD4 inhibitor JQ1, however the RNAsequencing was with 24 hour application of JQ1 on hiPSC cells. Therefore, an acute injection of JQ1 in mice prior to behavioral testing may better model the changes found within hiPSC cells.

3) QPCR validation for any of the RNA-sequencing targets.

4) Please explain why Fig1c vs Fig1d are normalized differently (H2A.Zac/H2A.Z vs H2A.Zac/GAPDH)

5) Figure 5: The sample size for Fig 5f is incorrect. The figure legend says n=10, but there are more data points shown than that.

6) There are no statistics reported in Figure 5. Also related to stats and figure 5, the methods section indicates that t-tests were used, but figure 5c, 5d and 5f require repeated measures ANOVA analysis.

7) There is no data showing that JQ1 affected BRD4 function in the animal experiments. That may be the only real weakness of the paper that could use addressing.

8) Please list the genes from the 698 manually curated neuronal genes. Please list the genes in the 8 subdivided categories.

9) Lastly, for the discussion, the manuscript is focused on neuronal genes and neuronal function of BRD4. But if BRD4 is expressed in other cell types, then this should be discussed and what it would mean to the overall interpretation of the work. Also, for the discussion, what do the authors make of the data in Figure 1d, in which over half of the SZ samples do not show elevated H2A.Z acetylation?

Reviewer #3 (Remarks to the Author):

In this manuscript the authors carry out a study that seeks to link hyperacetylation of histones H2A_Z and H4 in schizophrenia. This was done by analyzing histone acetylation in patient derived iPSCs using an established protein mass spectrometry approach and also then validated by western blotting in these cells and in postmortem tissue. Next the authors analyzed the interaction of BRD4 and H2A-Zac followed by RNA Seq of control vs JQ1 treated hiPSC-neurons. Lastly, the authors sought to show that JQ1 could improve schizophrenia like phenotypes in wild type mice.

Overall, the manuscript is an interesting combination of molecular and omic technologies to obtain a better understanding of schizophrenia. For the most part, the data is appropriately obtained with replicates and follow up validation. The major weakness of the manuscript, however, is Figure 5 where the authors claim to have improved SZ like phenotypes in wild type mice. There are significant differences in this figure with respect to the effect of JQ1 on amphetamine treatment, for example, but these behavioral effects do not come across as well developed and it is unclear if there are better ways to test this hypothesis. There is also a very large conceptual jump from the RNA Seq data analysis in prior figures to the behavioral analysis in Figure 5.

To improve this body of work there are two directions that the authors could go. One is to add more mouse model SZ data to better convince the reviewers of the potential of JQ1 for treatment of SZ. This is likely quite challenging and probably expensive.

The other possibility, and perhaps the better one, is to remove Figure 5 from the manuscript and instead carry out further molecular analyses of hiPSC-neurons in the presence or absence of JQ1, for example. Possible studies could include following up the gene expression changes with biochemical, imaging, or molecular approaches to better understand the pathways being affected by JQ1 and why H2A-Z and histone H4 might be important for SZ like phenotypes in hiPSC-neurons. This would likely result in a more compelling body of work providing a better molecular and mechanistic understanding of the epigenetic processes that might affect SZ

We apologize for the delay in resubmission, which reflects research disruptions, care-giving burdens, parental/companion losses and illnesses, and household/institutional/career moves associated with the ongoing COVID-19 pandemic across our team.

We wish to thank the reviewers for their thoughtful comments and suggestions, which we found to be extraordinarily helpful in strengthening our manuscript. We individually address each comment below in blue; our major revisions are summarized as:

- New data (Fig. S2, S3, S5b) to further strengthen observed biochemical and genomic links between BET family proteins and H2A.Zac;
- Removal of all rodent JQ1 manipulation data (previously Fig. 5);
- Revision of the text to more carefully reflect and interpret the data presented.

Reviewer #1 (Remarks to the Author):

In this report, Farrelly and colleagues characterize posttranslational modifications in human iPSC-derived neurons from individuals with schizophrenia. MS analysis reveals elevated acetylation level for several lysine residues of H2AZ in these neurons, and this finding was confirmed in postmortem human brain. The authors further show that bromodomains of BRD4 can bind polyacetylated H2AZ peptides, albeit weakly, in a hundreds-micromolar to millimolar range, and the second bromodomain can be crystallized in complex with an H2AZK4ac/K7ac/K11ac peptide. Finally, the authors show that JQ1 treatment results in changes in gene expression in neurons and improves sensorimotor gating behavior in mice.

This work describes novel findings in an exciting area relevant to human health, though the manuscript could be further strengthened by incorporating additional more convincing data. The major weakness from my point of view is that the three parts although exciting are not as connected as they should, i.e. (i) finding that H2AZ is hyperacetylated at some sites, (ii) BRD4 bromodomains bind acetylated H2AZ peptides, however very weakly in comparison to binding of these domains to other acetylated, even monoacetylated, histones. This obviously raises the question- is the weak association of BDs with H2AZ physiologically relevant? (iii) JQ treatment affects gene expression patterns, which is expected, granted JQ1 targets BRD4 BDs, but there is no evidence that H2AZ acetylation is involved here.

We very much appreciate the Reviewer's positive comments on our manuscript and constructive suggestions for additional experiments. We have done our best to respond to each of the Reviewer's points, as discussed below.

A few minor comments: binding affinity could be increased perhaps by testing linked bromodomains. Is N433A folded? A rationale to go after BRD4 could be strengthened by testing other known acetyllysine binding domains along with BRD4 BDs by pull-down experiments or peptide arrays.

1) We concur that the binding interactions presented in our initial submission between BRD4 and H2A.Zac are indeed "weaker," relatively speaking, in comparison to BRD4 binding with some other histone PTMs (e.g., H4K5acK8ac). This is likely due to the sequence motif of H2A.Z (see alignment below). The binding of BRD4 to H2A.Zac is comparable to that of the Twist peptide, in which binding

affinities of 0.6 and 3 mM were measured for BRD4 Bromo1 and Bromo2, respectively. (<https://www.sciencedirect.com/science/article/pii/S1535610814000427>).

Based upon sequence motif analyses, we believe that Ala5 and Asp8 may cause observed lower binding affinities. This might be because Ala5 (also Arg of TWIST) is less flexible in comparison to Gly6 of histone H4. Also, Asp8 is less optimal for BRD4 binding, owing to its negative charge (i.e., histone PTM “readers” typically prefer positively charged residues).

H2A.Z: AGG**KacA**AG**KacD**SGKAKTK
 TWIST: SPAQG**KacR**G**KacK**SA
 H4K5K8: SGRG**KacG**G**KacG**GLGK

Given the fact that multivalent interactions may indeed further prompt such binding, as suggested by the Reviewer, we performed ITC using the Bromo1-linker(GGS)₅-Bromo2 of BRD4 titrated with H2A.ZK4ac7ac11ac peptides cross-linked by a C-terminal cysteine (see **Figure S2b** below). Compared with Bromo2 and Bromo1 of BRD4, the extended Bromo1-linker-Bromo2 of BRD4 promotes further binding of BRD4 towards polyacetylated H2A.Z by 1.7- and 6.49-fold, respectively.

Figure S2

2) To confirm that the mutant (N433A) is folded, we have now tested this protein *vs.* its WT counterpart via Circular Dichroism (see **Figure S2a** above).

3) According to the sequence alignment of BET family members (see below), the binding pocket is relatively conserved. Therefore, we performed additional ITC assessments of other BET family members against H2A.Zac *vs.* unmodified peptides. As expected, BRD2_{Bromo1/2} and BRD3_{Bromo1/2} can also bind to polyacetylated H2A.Z, which is consistent with our sequence alignment analysis (see **Figure S2c-d** above). We have additionally toned down our interpretations in the manuscript regarding BRD4 specificity to now better reflect that H2A.Zac is a binding target of multiple BET family proteins, which can be inhibited by JQ1 to attenuate inappropriate SZ related gene expression.

4) In an attempt to connect BRD4 more specifically to SZ related gene expression, we first attempted ChIP-seq experiments in hiPSC neurons (control *vs.* SZ) for BRD4 and H2A.Zac. However, we were unable to obtain high quality BRD4 ChIP-seq data (testing multiple supposed ChIP grade antibodies) to be used in comparisons against H2A.Zac ChIP-seq results. We then attempted to perform CUT&RUN followed by sequencing in hiPSC neurons (control *vs.* SZ) for BRD4 and H2A.Zac. Unfortunately, this attempt similarly did not yield high quality BRD4 ChIP-seq data.

During revisions, a manuscript was published presenting high quality BRD4 ChIP-seq data in human cells (HeLa; <https://www.nature.com/articles/s41467-020-17503-y>; GSA151038). As such, we repeated ChIP-seq for H2A.Zac in HeLa cells so as to be able to compare BRD4 *vs.* H2A.Zac enrichment patterns genome-wide in a human cell-type. Given that both proteins/marks are known to enrich at genic loci,

Figure S3

specifically within promoters, counts of reads mapping to promoter regions for H2A.Zac and BRD4 were obtained using featureCounts. Raw counts were then converted to RPKM, and the corrgram R package was implemented to compute spearman rank correlations between the coverage of promoter regions for BRD4 and H2A.Zac datasets. In doing so, we found that BRD4 and H2A.Zac enrichments are highly correlated genome-wide (0.88; $= <2e-16$) – see **Figure S3a-b** above – as predicted based upon our biophysical assessments. Based upon these results, we next performed BRD4 manipulations (via JQ1 treatments) in HeLa cells to assess whether genes co-enriched for BRD4 and H2A.Zac are affected in their expression by BRD4 inhibition. As predicted, these co-enriched loci were found to be repressed by such manipulations (see **Figure S3c** above).

In a final attempt to more directly link BRD4 binding to H2A.Zac mediated gene expression in human cells, we generated stable HeLa cell lines expressing either FLAG tagged wildtype BRD4 or mutant BRD4 (N433A), with the intention being to demonstrate whether an inability to recruit BRD4 to H2A.Zac marked genes specifically (as in the case of BRD4-N433A) may perturb the expression of these co-enriched loci. Unfortunately, however, exogenous overexpression of BRD4 (WT and mutant) resulted in the presence of massively enlarged nuclei (**see below**) – likely owing to chromatin de-compaction, as suggested by the literature – making interpretations of gene expression outputs highly confounded. As such, we chose to exclude these data from the revised manuscript.

Given all of the findings discussed above, along with our observation that poly-acetylated H4 is similarly induced in SZ neurons and postmortem brain (**Figure S1a-e**) vs. H2A.Zac, we have limited our interpretations in the revised manuscript to more definitively indicate a) that histone acetylation (both combinatorial H2A.Zac and H4ac) are inappropriately induced in SZ neurons/brain, b) that BRD4 binds to/co-enriches genome-wide with combinatorial H2A.Zac, similar to that of H4ac, and c) that inhibition of BRD4 interactions with acetylated histones via BET family inhibition in neurons effectively ameliorates SZ related gene expression. To date, other groups have not yet published similar findings, and we continue to feel strongly that these results will be of broad interest to both the neuroscience and chromatin fields.

Reviewer #2 (Remarks to the Author):

This manuscript identifies novel histone acetylation patterns of the H2A.Z variant histone proteins (e.g. H2A.Z.1/2-K4acK7acK11Ac) within human-induced pluripotent stem cell derived (hiPSC) forebrain neurons from individuals with schizophrenia (SZ) using mass spectrometry. Following this, the authors confirm that one of the bromodomains of the bromodomain containing protein BRD4 (BRD4Bromo2) has high affinity for, and interacts with, H2A.Z.K4acK7acK11ac using isothermal titration calorimetry. RNA-Sequencing analysis on hiPSC-derived neurons from SZ cases and controls that were treated with a BRD4 inhibitor, JQ1, reveals that BRD4 inhibition alters a subset of SZ-related gene expression changes. Lastly, chronic injections of JQ1 blocked amphetamine-induced hyperlocomotion in mice, and increased prepulse inhibition in a different group of JQ1-treated mice. Overall, these studies demonstrate that BRD4 inhibition ameliorates transcriptional and behavioral changes typically seen in SZ cases and also that BRD4 binds acetylation sites that are hyperacetylated in hiPSCs-derived from SZ samples. These results significantly contribute to our understanding of how the epigenome is altered in SZ, how BRD4 may be interacting with hyperacetylated lysines, and the potential use of BRD4 inhibition to alter sensorimotor phenotypes.

We very much thank the Reviewer for their positive comments on our paper and their appreciation of the “significant” contributions that our work provides “to [the field’s] understanding of how the epigenome is altered in SZ.”

Some additional experiments are suggested, but not necessary, prior to publishing this manuscript. Other minor issues mentioned below as well.

1) In vitro ITC studies demonstrate that 1) BRD4 interacts with H2A.Z.K4acK7acK11Ac and 2) JQ1 abolishes binding of BRD4Bromo2 to tri-acetylated H2A.Z. In addition, RNA-sequencing studies find that application of the BRD4 inhibitor alters SZ-related gene expression in hiPSC-SZ samples. However, it may serve to benefit the conclusions made from this work by examining whether the mice that exhibited improvements in SZ-related behavioral impairments have decreased binding of BRD4 at H2AZ.K4acK7acK11Ac, or similar gene expression changes found from RNA-sequencing within the prefrontal cortex.

Based upon suggestions from Reviewer #3 (see below) and discussions with the *Nature Communications* Editorial staff, we have removed all rodent behavioral analyses (previous Figure 5)

from the revised version of our manuscript. Doing so has allowed us to better focus our manuscript exclusively on human findings, which are more relevant to the disease in question (i.e., SZ).

2) Behavioral studies were performed with chronic (7 day) treatment of the BRD4 inhibitor JQ1, however the RNAsequencing was with 24 hour application of JQ1 on hiPSC cells. Therefore, an acute injection of JQ1 in mice prior to behavioral testing may better model the changes found within hiPSC cells.

Please see our response to comment #1 above.

3) qPCR validation for any of the RNA-sequencing targets.

We now provide qPCR validations of select target genes elucidated from our RNA-seq data (from differential analyses) comparing control vs. SZ hiPSC neuronal samples +/- JQ1. These are now presented in **Figure S5b** (see below) and are used to further support our subsequent computational analyses aimed at elucidating the additive vs. synergistic effects of JQ1 treatments in SZ neurons.

Figure S5b

4) Please explain why Fig1c vs Fig1d are normalized differently (H2A.Zac/H2A.Z vs H2A.Zac/GAPDH)

Despite our best efforts, we were unable to obtain high quality western blotting data using the H2A.Z (unmodified) antibody in human postmortem brain. As such, we chose to use GAPDH as a loading control for postmortem comparisons, because its total levels were unaffected by psychiatric diagnosis. This is now discussed in the Methods.

5) *Figure 5: The sample size for Fig 5f is incorrect. The figure legend says n=10, but there are more data points shown than that.*

Please see our response to comment #1 above.

6) *There are no statistics reported in Figure 5. Also related to stats and figure 5, the methods section indicates that t-tests were used, but figure 5c, 5d and 5f require repeated measures ANOVA analysis.*

Please see our response to comment #1 above.

7) *There is no data showing that JQ1 affected BRD4 function in the animal experiments. That may be the only real weakness of the paper that could use addressing.*

Please see our response to comment #1 above.

8) *Please list the genes from the 698 manually curated neuronal genes. Please list the genes in the 8 subdivided categories.*

We recently published a detailed method for evaluating synergy-driving gene expression, including all 698 manually curated neuronal genes as eight subdivided categories, which are found in Supplementary Data 1 of Schrode et al, 2021 (https://static-content.springer.com/esm/art%3A10.1038%2Fs41596-020-00436-7/MediaObjects/41596_2020_436_MOESM2_ESM.zip). We now cite this manuscript in the methods and results.

Schrode, N., Seah, C., Deans, P.J.M. et al. Analysis framework and experimental design for evaluating synergy-driving gene expression. *Nat Protoc* 16, 812–840 (2021). <https://doi.org/10.1038/s41596-020-00436-7>.

9) *Lastly, for the discussion, the manuscript is focused on neuronal genes and neuronal function of BRD4. But if BRD4 is expressed in other cell types, then this should be discussed and what it would mean to the overall interpretation of the work. Also, for the discussion, what do the authors make of the data in Figure 1d, in which over half of the SZ samples do not show elevated H2A.Z acetylation?*

We agree that this is indeed an interesting question, especially as it relates to the impacts of JQ1 administration when given systemically (e.g., behavioral effects observed may be due to glial regulation of BET family proteins instead of neuronal BET activities). By removing the JQ1 manipulation data in rodents from the resubmission, we were able to focus our manuscript exclusively on the impacts of JQ1 administration in human neurons in the context of SZ diagnosis, thereby alleviating the need to discuss such potential caveats.

With respect to the variability observed in Fig. 1d – unfortunately, besides clinical diagnosis and a few other co-variates (race, age, PMI, etc.), we only have a very limited case histories for each of the subjects examined in this study. As such, we do not have information about drug histories (therapeutic or otherwise), range of clinical symptoms observed (as SZ is highly heterogenous), genotypes, etc., all of which may influence these types of molecular outputs. However, given that even with such modest Ns of postmortem samples we were able to validate effects observed in our hiPSC cohorts makes us extremely confident in the robustness of observed effects.

Reviewer #3 (Remarks to the Author):

In this manuscript the authors carry out a study that seeks to link hyperacetylation of histones H2A_Z and H4 in schizophrenia. This was done by analyzing histone acetylation in patient derived iPSCs using an established protein mass spectrometry approach and also then validated by western blotting in these cells and in postmortem tissue. Next the authors analyzed the interaction of BRD4 and H2A-Zac followed by RNA Seq of control vs JQ1 treated hiPSC-neurons. Lastly, the authors sought to show that JQ1 could improve schizophrenia like phenotypes in wild type mice.

Overall, the manuscript is an interesting combination of molecular and omic technologies to obtain a better understanding of schizophrenia. For the most part, the data is appropriately obtained with replicates and follow up validation. The major weakness of the manuscript, however, is Figure 5 where the authors claim to have improved SZ like phenotypes in wild type mice. There are significant differences in this figure with respect to the effect of JQ1 on amphetamine treatment, for example, but these behavioral effects do not come across as well developed and it is unclear if there are better ways to test this hypothesis. There is also a very large conceptual jump from the RNA Seq data analysis in prior figures to the behavioral analysis in Figure 5.

To improve this body of work there are two directions that the authors could go. One is to add more mouse model SZ data to better convince the reviewers of the potential of JQ1 for treatment of SZ. This is likely quite challenging and probably expensive.

The other possibility, and perhaps the better one, is to remove Figure 5 from the manuscript and instead carry out further molecular analyses of hiPSC-neurons in the presence or absence of JQ1, for example. Possible studies could include following up the gene expression changes with biochemical, imaging, or molecular approaches to better understand the pathways being affected by JQ1 and why H2A-Z and histone H4 might be important for SZ like phenotypes in hiPSC-neurons. This would likely result in a more compelling body of work providing a better molecular and mechanistic understanding of the epigenetic processes that might affect SZ

We thank Reviewer 3 for their positive comments on our manuscript. Given Reviewer 2's and 3's concerns over the behavioral data previously presented in Figure 5 (and following discussions with the Editorial staff at *Nature Communications*), we agree that it is best to remove these data from the current manuscript and reserve them for inclusion in future publications. Removing the rodent data also allowed us to focus our paper exclusively on the human findings, which are indeed more relevant to the disease in question (i.e., SZ). In addition, we have added further biochemical and molecular analyses to further bolster connections between BRD4 and H2A.Zac (e.g., further ITC assessments of multivalent

interactions between BRD4 and H2A.Zac, binding assessments between other BET family proteins and H2A.Zac, genome-wide comparisons of BRD4 and H2A.Zac enrichment in human cells, etc.).

As discussed above in response to Reviewer 1, given our updated molecular and biochemical analyses (including our various attempts to more definitively link altered H2A.Zac in SZ neurons to BRD4 recruitment, many of which unfortunately did not yield high quality data owing to technical issues, such as antibody quality), along with our observation that poly-acetylated H4 is similarly induced in SZ neurons and postmortem brain (**Figure S1a-e**), we have now limited our interpretations in the revised manuscript to more specifically indicate a) that histone acetylation (both combinatorial H2A.Zac and H4ac) are inappropriately induced in SZ neurons/brain, b) that BRD4 binds to/co-enriches genome-wide with combinatorial H2A.Zac, similar to that of H4ac, in human cells and c) that inhibition of BRD4 interactions with acetylated histones via BET family inhibition in neurons effectively ameliorates SZ related gene expression. To date, no other groups have yet published similar findings, and we continue to feel strongly that these results will be of broad interest to both the neuroscience and chromatin communities.

REVIEWERS' COMMENTS

Reviewer #1 (Remarks to the Author):

The authors have adequately addressed my previous comments.

Reviewer #2 (Remarks to the Author):

The revised manuscript is much improved, especially with removal of the rodent studies in the previous Figure 5. That allowed many weaknesses in the study to be removed and a more clear focus on H2AZ acetylation, BRD4 interaction, and cell culture work. The interpretation has been adjusted appropriately as well. Overall, the authors have done a great job revising the study over the past year and a half resulting in a significant contribution to our understanding of histone variant acetylation, BRD4 activity, and how this activity may relate to SZ.

Reviewer #3 (Remarks to the Author):

Farrelly et al present a revised manuscript with the primary change being the removal of the data in Figure 5 where behavioral analysis was presented, but quite underdeveloped. The removal of this aspect of the work has improved manuscript by being more focused on the BRD4 and H2A.Z. The revised manuscript, however, does not have a great deal of additional data in support of this. There is the addition of some supplemental figures, but in general the manuscript has gone from 5 main figures to 4 figures. Essentially it is a good not great revised version of the manuscript. That being said, the topic is important and the data in the manuscript is strong, I support publication of the revised manuscript in its current form.

We would like to thank the Reviewers for their thoughtful comments and suggestions throughout the entire review process, which we found to be extraordinarily helpful in strengthening our manuscript. Below, we respond to the Reviewers final comments about our paper.

Reviewer #1 (Remarks to the Author):

The authors have adequately addressed my previous comments.

We very much appreciate the Reviewer's helpful critiques throughout the review process and are pleased to know that we have fully addressed their previous comments.

Reviewer #2 (Remarks to the Author):

The revised manuscript is much improved, especially with removal of the rodent studies in the previous Figure 5. That allowed many weaknesses in the study to be removed and a more clear focus on H2AZ acetylation, BRD4 interaction, and cell culture work. The interpretation has been adjusted appropriately as well. Overall, the authors have done a great job revising the study over the past year and a half resulting in a significant contribution to our understanding of histone variant acetylation, BRD4 activity, and how this activity may relate to SZ.

We again thank the Reviewer for their positive comments on our paper and their appreciation of the significant contributions that our work provides to the field's understanding of how the epigenome is altered in SZ.

Reviewer #3 (Remarks to the Author):

Farrelly et al present a revised manuscript with the primary change being the removal of the data in Figure 5 where behavioral analysis was presented, but quite underdeveloped. The removal of this aspect of the work has improved manuscript by being more focused on the BRD4 and H2A.Z. The revised manuscript, however, does not have a great deal of additional data in support of this. There is the addition of some supplemental figures, but in general the manuscript has gone from 5 main figures to 4 figures. Essentially it is a good not great revised version of the manuscript. That being said, the topic is important and the data in the manuscript is strong, I support publication of the revised manuscript in its current form.

We appreciate the Reviewer's sentiment that the topic of our manuscript is important, that the data presented are strong and that our revised paper should now be published.